# Oxytocin Deficiency in Childhood and Adolescence: Clinical Features, Diagnostic Challenges and Therapeutic Perspectives

**DOI:** 10.3390/cimb47120982

**Published:** 2025-11-25

**Authors:** Roberto Paparella, Arianna Bei, Irene Bernabei, Cinzia Fiorentini, Norma Iafrate, Roberta Lucibello, Lavinia Marchetti, Francesca Pastore, Vittorio Maglione, Marcello Niceta, Marco Fiore, Brunella Caronti, Mario Vitali, Ida Pucarelli, Luigi Tarani

**Affiliations:** 1Department of Maternal Infantile and Urological Sciences, Sapienza University of Rome, 00185 Rome, Italy; 2Molecular Genetics and Functional Genomics, Ospedale Pediatrico Bambino Gesù, IRCCS (Istituto di Ricovero e Cura a Carattere Scientifico), 00165 Rome, Italy; 3Institute of Biochemistry and Cell Biology (IBBC-CNR), c/o Department of Sensory Organs, Sapienza University of Rome, 00185 Rome, Italy; 4Department of Human Neurosciences, Sapienza University of Rome, 00185 Rome, Italy; 5ASUR Marche, AV4, 60131 Ancona, Italy

**Keywords:** oxytocin, hypothalamic-pituitary disorders, Prader–Willi syndrome, pediatrics, clinical trials

## Abstract

Oxytocin (OXT), traditionally linked to reproductive physiology, is now recognized as an important regulator of metabolic, skeletal, and socio-emotional processes. In children and adolescents, oxytocin deficiency (OXT-D) represents a significant but frequently underdiagnosed neuroendocrine disturbance, particularly in hypothalamic–pituitary disorders and syndromic conditions such as Prader–Willi and Schaaf–Yang. Experimental and clinical evidence suggests that OXT-D may contribute to altered appetite regulation, reduced energy expenditure, impaired bone health, and socio-emotional vulnerability, even when other pituitary axes are adequately replaced. Diagnostic evaluation remains challenging due to OXT’s short half-life, pulsatile secretion, and the limited reliability of current assay platforms, which restrict the clinical utility of peripheral measurements or dynamic testing in pediatric practice. Intranasal OXT—the most extensively studied therapeutic approach—shows good short-term tolerability and context-dependent behavioral benefits, though long-term efficacy and safety remain insufficiently defined. Advancing the field will require standardized diagnostic criteria, more reliable biomarkers, and precision-medicine strategies accounting for developmental stage and genetic background. This review summarizes current knowledge on pediatric OXT-D and highlights priorities for future translational and clinical research.

## 1. Introduction

Oxytocin (OXT), originally identified as a neurohypophyseal peptide mediating uterine contractions and milk ejection, was long regarded primarily as a reproductive hormone. Advances in neuroendocrine and molecular research have since demonstrated that OXT also regulates key metabolic, skeletal, and socio-emotional processes, broadening its relevance far beyond parturition and lactation [1].

The development of OXT- and receptor-deficient animals was pivotal in this shift, revealing profound alterations in thermoregulation, body composition, and energy expenditure even in the absence of hyperphagia [2,3], and identifying OXT as an important modulator of metabolic and neuromuscular physiology [4]. Clinical observations parallel experimental findings. Children and adolescents with impaired oxytocinergic function—whether due to congenital syndromes such as Prader–Willi (PWS) or Schaaf–Yang (SYS), SIM1 deficiency, hypothalamic–pituitary lesions, or early-life adversity—frequently display normophagic obesity, reduced lean mass, and socio-emotional difficulties despite adequate replacement of other pituitary hormones [4,5]. Moreover, OXT has been recognized as a pleiotropic neuropeptide implicated in neurodevelopment, social behavior, and emotional regulation [6]—domains particularly relevant during childhood and adolescence, when neural circuits governing social cognition and stress reactivity undergo rapid maturation [7,8].

Given this heterogeneity, the term OXT-D in this review is used in a clinically meaningful but structured manner, encompassing three partially overlapping mechanisms: (1) quantitative OXT-D, referring to reduced hormone secretion, typically from hypothalamic–pituitary injury; (2) functional OXT-D, indicating impaired receptor availability or signaling efficiency, as observed in OXT receptor (OXTR) polymorphisms, promoter methylation, or downstream pathway defects; (3) network dysregulation, reflecting imbalances within OXT–vasopressin (AVP) circuits or altered integration in limbic and hypothalamic pathways.

Despite growing recognition of OXT-D as a distinct pediatric neuroendocrine entity, translation into clinical practice remains limited. Accurate measurement of endogenous OXT is technically challenging due to its short half-life, pulsatile secretion, and assay variability, and interventional pediatric studies are still scarce [9,10]. These barriers complicate both diagnosis and treatment development. This review synthesizes current evidence on pediatric OXT-D and outlines emerging diagnostic and therapeutic perspectives.

## 2. Physiology of Oxytocin and Vasopressin in Development

### 2.1. Sites of Synthesis and Release

OXT and AVP are structurally related nonapeptides with high sequence homology and partially overlapping functions [11,12]. Their genes lie in close proximity on chromosome 20p13. Both hormones are synthesized predominantly by magnocellular neurons of the paraventricular and supraoptic nuclei, whose axons project to the posterior pituitary. Here, OXT and AVP are packaged into neurosecretory vesicles and released into the systemic circulation in response to defined physiological stimuli, mediating their classical endocrine actions [13,14,15].

Within secretory granules, OXT and AVP are bound to their respective carrier proteins—neurophysin I and neurophysin II—in a 1:1 ratio [16]. After systemic release, they cannot re-enter the central nervous system (CNS) because of the blood–brain barrier; however, both peptides are also released centrally through dendritic and axonal mechanisms in the hypothalamus and other brain regions, enabling autocrine and paracrine modulation of local circuits [17,18,19].

Following peripheral secretion, OXT has a short plasma half-life of approximately 3–5 min due to rapid enzymatic degradation by oxytocinases, whereas clearance in cerebrospinal fluid (CSF) is slower [20]. Central OXT concentrations in the supraoptic nucleus and CSF are markedly higher than in plasma, reflecting the compartmentalized regulation of the oxytocinergic system [13,17,21].

Beyond hypothalamic production, extra-hypothalamic OXT synthesis has been described in bone, kidney, adipose tissue, myocardium, and the gastrointestinal tract, where it exerts autocrine and paracrine actions that contribute to bone remodeling, lipid metabolism, energy expenditure, and gastrointestinal motility [18].

### 2.2. Receptors and Cross-Reactivity

The biological actions of OXT and AVP are mediated by distinct but evolutionarily related G protein–coupled receptors (GPCRs). OXT primarily activates the OXTR, whereas AVP signals through the V1aR, V1bR, and V2R subtypes [13,22]. The *OXTR* gene is located on chromosome 3p26.2, and the receptor can form heterodimers with other GPCRs—including ghrelin (GHS-R1a) and serotonin (5-HT2A, 5-HT2C) receptors—suggesting broader neuromodulatory interactions [23,24]. Receptor expression is dynamically regulated by hormonal, inflammatory, and epigenetic influences; estrogens, IL-6, IL-1, and promoter methylation can modulate *OXTR* transcription and contribute to variability in individual responsiveness to OXT [25,26].

Partial cross-reactivity exists between the OXT and AVP systems: OXT can weakly stimulate V1aR, and AVP may bind OXTR at high concentrations. Their distinct physiological roles derive from differences in receptor-binding affinity, where even single amino acid substitutions significantly alter ligand selectivity [27,28,29]. These differences may have clinical implications—for example, in conditions such as diabetes insipidus, where receptor sensitivity or ligand specificity may be altered [30].

OXTR and AVP receptors are widely expressed across the central and peripheral nervous systems, including the hippocampus, amygdala, medial prefrontal cortex, bed nucleus of the stria terminalis, as well as kidney, heart, pancreas, reproductive tissues, skeletal muscle, adipose tissue, and bone [22]. Receptor localization along the vagus nerve offers a mechanistic pathway through which peripheral OXT signals can influence central circuits, highlighting its integrative role in gut–heart–brain communication [31,32].

Functional studies also suggest that OXT may modulate prolactin, gonadotropin, and adrenocorticotropic hormone release via short portal connections between the posterior and anterior pituitary, although OXTR expression within the human adenohypophysis remains uncertain [22,33,34].

### 2.3. Developmental Aspects

The ontogeny of OXT and AVP signaling is highly dynamic. During prenatal and early postnatal life, OXTR expression peaks in brain regions involved in sensory integration and social learning, supporting mother–infant bonding and the maturation of neural circuits underlying attachment, social recognition, and stress regulation [35,36,37]. Developmental stage, sex, and genetic background shape the expression patterns of OXTR and AVP receptors, contributing to inter-individual differences in social behavior and susceptibility to neurodevelopmental disorders [35]. Early-life environmental exposures—including variations in maternal care, psychosocial stress, or adversity—can induce durable epigenetic modifications in OXT and AVP pathways. These programming effects may persist across development, influencing socio-emotional processing and vulnerability to psychiatric and behavioral conditions [19,24,38].

### 2.4. Peripheral Versus Central Actions in Pediatric Physiology

OXT and AVP have well-established peripheral roles: OXT is essential for uterine contractions and milk ejection [39], while AVP regulates fluid balance and vascular tone [14,30,40]. Increasing attention now focuses on their central actions, which are crucial for social behavior, cognition, and emotional regulation [Figure 1].

In childhood, OXT and AVP contribute to social learning, bonding, and the processing of social cues, with dysregulation implicated in ASD, Prader–Willi syndrome, and other neurodevelopmental conditions involving hypothalamic–pituitary dysfunction [4,35,41,42,43]. Peripherally, OXT influences metabolic function, bone turnover, and muscle regeneration. Evidence from animal and human studies indicates that OXT enhances insulin sensitivity, promotes lipolysis, stimulates osteoblast activity, and supports muscle maintenance—processes particularly relevant during growth and puberty [11,16,18,39].

Together, these peptides act as multifunctional regulators whose coordinated central and peripheral effects are essential for metabolic homeostasis and socio-emotional development. Disruption of these pathways during critical windows may contribute to endocrine, metabolic, and neuropsychiatric disorders in pediatric populations.

**Figure 1 cimb-47-00982-f001:**
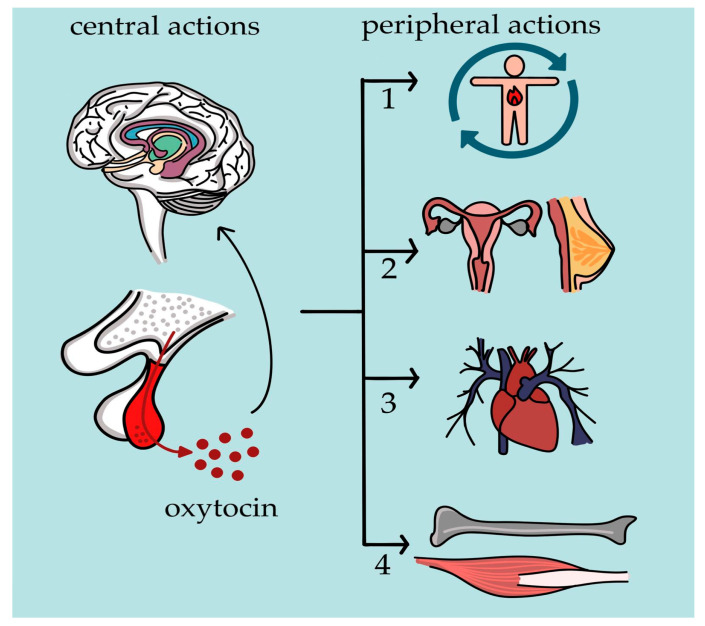
**Central and peripheral actions of oxytocin (OXT).** At the central level, OXT acts on the central nervous system and limbic circuits, including the amygdala, hippocampus, and hypothalamus. OXT is involved in the modulation of social behavior—such as attachment, trust, empathy, and social cohesion—as well as in the regulation of stress and anxiety responses, sexual and reproductive behavior, and feeding and appetite control [14]. Peripherally, OXT modulates uterine contractility, milk ejection, and ovarian and testicular function (2). It also exerts protective effects on the cardiovascular system (3), regulates glucose and lipid metabolism (1), and promotes anabolic effects on muscle and bone tissue (4) [39,44].

## 3. Etiology of Oxytocin Deficiency in Pediatric Populations

The origins of OXT-D in childhood and adolescence are multifactorial and arise from the interaction of structural, genetic, and environmental factors. These etiologies align with the three dimensions of OXT-D described in the Introduction: reduced hormone secretion, impaired receptor or signaling function, and broader dysregulation of OXT–AVP networks. Key contributors include congenital or acquired hypothalamic–pituitary disorders and genetic or epigenetic alterations affecting OXTR. Environmental, psychosocial, and nutritional influences may further modify oxytocin signaling. Understanding these pathways is essential for explaining the heterogeneity of clinical features and for guiding diagnostic and therapeutic strategies in pediatric patients.

### 3.1. Hypothalamic-Pituitary Axis

#### 3.1.1. Congenital Causes

PWS is the best-characterized congenital cause of quantitative OXT-D [14,45]. Loss of paternally expressed genes on 15q11–q13 results in global hypothalamic dysfunction with neuroendocrine and behavioral consequences [43,46]. Both human and animal studies show impaired development and activity of oxytocinergic neurons, with reduced OXT tone and abnormal hypothalamic connectivity. These alterations contribute to the characteristic features of PWS, including neonatal hypotonia, hyperphagia with early-onset obesity, thermoregulatory instability, and socio-emotional difficulties [4,47,48].

Schaaf–Yang syndrome (SYS), caused by pathogenic *MAGEL2* variants, shares many molecular and phenotypic features with PWS. Defects in OXT synthesis and release are increasingly recognized as a central mechanism underlying their overlapping metabolic and behavioral profiles [49,50,51].

Additional neurodevelopmental conditions—including Williams syndrome, Fragile X syndrome, and selected forms of ASD—have also been linked to abnormalities in OXT or AVP pathways [16]. Although mechanisms are less clearly defined, these findings support a role for altered OXT signaling as a convergent pathway contributing to social and cognitive impairments in early development. In ASD, reduced circulating OXT levels and *OXTR* gene polymorphisms or epigenetic modifications have been associated with increased susceptibility and social communication deficits [52,53].

#### 3.1.2. Acquired Causes

Acquired OXT-D most often results from structural or functional damage to the hypothalamic–pituitary region [9]. Craniopharyngioma is the prototypical pediatric cause: this benign but locally invasive tumor frequently involves the sellar and suprasellar areas, compromising OXT-producing neurons in the paraventricular and supraoptic nuclei. Both the mass effect and its treatments—particularly surgery and radiotherapy—may impair OXT synthesis or release [54,55,56]. Affected patients may show normal basal OXT levels but blunted responses to physiological stimuli, a pattern associated with anxiety, increased adiposity, and social difficulties despite optimal replacement of other pituitary hormones [57,58].

Other hypothalamic lesions, including germ cell tumors, Langerhans cell histiocytosis, and gliomas, can similarly disrupt oxytocinergic pathways [5,59,60]. Autoimmune or inflammatory forms of hypophysitis, such as lymphocytic or IgG4-related subtypes, may also impair posterior pituitary function and secondarily affect OXT secretion, although direct evidence of OXT-D in these conditions is still lacking [61].

#### 3.1.3. Secondary Deficiency and Neurohypophyseal Dysfunction

Because of their anatomical proximity, secondary OXT-D can accompany broader neurohypophyseal disorders such as panhypopituitarism or central diabetes insipidus (CDI) [1,54]. However, the clinical overlap between AVP and OXT deficits is only partial. AVP loss typically presents with the polyuria and polydipsia of CDI [62], whereas OXT deficiency manifests more subtly through alterations in emotional regulation, feeding behavior, and social cognition. Not all patients with CDI show features attributable to OXT-D, and conversely, children with hypothalamic injury from craniopharyngioma may display social or behavioral abnormalities even without CDI [55]. These differences emphasize that OXT status cannot be inferred from AVP function alone and requires dedicated evaluation.

### 3.2. Genetic and Epigenetic Alterations of the Oxytocin Receptor

Genetic and epigenetic variability within the *OXTR* gene is a key contributor to functional OXT-D and to individual differences in oxytocin signaling [63]. Several single-nucleotide polymorphisms—particularly rs53576, rs2254298, and rs237885—affect receptor expression or downstream signaling efficiency, as summarized by Kohlhoff et al. [64]. These variants have been linked to differences in social cognition, stress responsiveness, emotional regulation, and attachment, as well as to conditions such as depression, anxiety, alcohol misuse, borderline personality traits, callous–unemotional profiles, and ASD [65,66].

Such polymorphisms may impair *OXTR* transcription, mRNA stability, or receptor coupling, thereby reducing OXT biological efficacy. Epigenetic factors further modulate this vulnerability: early-life stress, maternal deprivation, and malnutrition can induce *OXTR* promoter methylation or histone modifications, leading to reduced receptor availability and diminished social sensitivity [29,67,68,69]. These gene–environment interactions reinforce the concept of functional OXT-D as a convergence of genetic predisposition and environmental influence.

In pediatric and adolescent populations, altered OXTR signaling may contribute to the heterogeneity of disorders involving hypothalamic–pituitary dysfunction, ASD, and PWS, where limited receptor availability exacerbates metabolic and socio-behavioral difficulties. Early environmental insults, including prenatal alcohol exposure, can further disrupt neurodevelopment during critical periods of oxytocinergic plasticity [70]. Overall, genetic and epigenetic alterations of *OXTR* represent central mechanisms linking early-life adversity, impaired receptor signaling, and complex clinical phenotypes in pediatric endocrinology [64,71].

### 3.3. Nutritional Factors

Nutritional deprivation during fetal or early postnatal life represents an important and often underrecognized contributor to impaired OXT signaling [72,73]. Experimental and clinical evidence indicates that prenatal undernutrition has long-lasting effects on hypothalamic neuropeptide systems, including OXT and OXTR. In a developmental rat model, maternal caloric restriction disrupted the maturation of oxytocinergic neurons and altered OXT and OXTR expression across early life stages, suggesting that energy deficiency perturbs metabolic programming during critical developmental windows [72]. These effects may be mediated by epigenetic remodeling, altered leptin–cortisol balance, and impaired neuronal development. Under physiological conditions, leptin activates OXT-producing neurons in the paraventricular nucleus; prenatal nutrient deprivation can disrupt this interaction, promoting leptin resistance and increasing long-term risk of obesity [74,75]. Dysregulated OXT signaling during early life may therefore impair satiety, energy expenditure, and socio-emotional adaptation, consistent with features of partial OXT-D.

Comparable alterations have been described in humans under chronic energy deprivation, including anorexia nervosa and low-energy availability in athletes. These conditions are characterized by reduced cerebrospinal and peripheral OXT levels, which correlate with low leptin, decreased resting energy expenditure, reduced bone mineral density, and impaired socio-emotional functioning [76,77,78]. Although OXT suppression may initially serve as an adaptive energy-conserving response, prolonged restriction can contribute to the clinical phenotype of functional OXT-D [16]. Importantly, OXT levels tend to normalize with weight restoration, indicating reversibility [77,79,80,81].

In summary, prenatal and postnatal malnutrition can induce a developmental form of OXT-D through hypothalamic reprogramming and altered receptor dynamics. Nutritional restriction later in life similarly suppresses oxytocinergic activity, affecting metabolic, cognitive, and socio-emotional health. Etiological contributors to OXT-D are summarized in Table 1.

## 4. Clinical Manifestations of Oxytocin Deficiency in Pediatrics

OXT-D in pediatric populations presents as a multisystem disturbance involving metabolic, neurobehavioral, psychiatric, endocrine, and growth-related domains. The phenotype depends on timing, severity, and etiology of disruption within oxytocinergic pathways, spanning congenital neurodevelopmental syndromes to acquired hypothalamic–pituitary injuries [4,10].

### 4.1. Metabolic Features

Metabolic abnormalities in pediatric OXT-D reflect both reduced hormone availability—particularly in hypothalamic lesions—and impaired downstream signaling, affecting appetite regulation, energy expenditure, and body composition [39]. Hyperphagia and poor satiety are characteristic in congenital forms such as PWS and SYS and in acquired hypothalamic dysfunction. OXT acts within hypothalamic and mesolimbic circuits to regulate homeostatic and hedonic feeding [82,83]; deficiency promotes increased adiposity, reduced lean mass, and altered lipid metabolism even without marked hyperphagia. Inflammatory biomarkers such as lipocalin-2 have been linked to metabolic dysregulation in pediatric endocrinopathies, a context in which impaired OXT signaling may contribute to adverse body composition [84].

Animal models with OXT or OXTR knockout consistently show late-onset obesity, reduced thermogenesis, and lower energy expenditure, highlighting OXT’s role in metabolic homeostasis [3,49]. Peripherally, OXT facilitates glucose uptake, lipolysis, and insulin sensitivity, supporting preservation of lean mass and limiting fat accumulation [85]. In humans, circulating OXT correlates positively with lean mass and bone mineral density and inversely with central adiposity [86]. Children with OXT-D may therefore present with normophagic or hypophagic obesity driven primarily by reduced energy utilization rather than increased intake, with potential implications for growth and pubertal development [76,87]. Oxidative stress, a common feature of various neuroendocrine and genetic disorders, may further exacerbate metabolic vulnerability in states of impaired OXT signaling [88].

OXT also contributes to bone homeostasis by stimulating osteoblast differentiation and inhibiting adipogenesis within the marrow environment [87,89,90,91]. Knockout models develop reduced bone mineral density, impaired bone geometry, and sarcopenia, some of which improve with exogenous OXT replacement [92]. In females, these anabolic effects appear partially estrogen-dependent [93,94].

Overall, OXT acts as an important metabolic and musculoskeletal regulator during childhood and adolescence. Persistent OXT-D may contribute to sarcopenic obesity, reduced bone accrual, and long-term metabolic fragility. However, most mechanistic evidence comes from adult or young-adult studies; pediatric-specific data remain limited and should be interpreted with caution.

### 4.2. Neurobehavioral and Psychiatric Features

OXT is a key regulator of social cognition, emotional modulation, and stress responsivity. Disruption of oxytocinergic signaling during critical developmental windows alters the maturation of prefrontal–limbic circuits, contributing to difficulties in empathy, attachment, emotion perception, and sleep–wake regulation [95,96,97]. Altered neurotrophin signaling may further compound these socio-emotional deficits in neurodevelopmental conditions [98,99].

Growing evidence links OXT pathway dysfunction to several neurodevelopmental and psychiatric disorders, largely reflecting functional OXT-D arising from impaired OXTR signaling or disrupted downstream circuit integration. Genetic and epigenetic alterations of the *OXTR* gene—such as single nucleotide polymorphisms and promoter hypermethylation—have been associated with increased vulnerability to ASD, attention-deficit/hyperactivity disorder, and conduct disorder [53,65,100,101]. Reduced plasma OXT levels correlate with impairments in attention, empathy, and prosocial engagement; in ASD, diminished OXTR expression and atypical activation of amygdala–insular networks contribute to altered social cognition [102,103,104].

Beyond neurodevelopmental conditions, OXT dysregulation has been implicated in schizophrenia and in mood and anxiety disorders. Intranasal OXT has shown variable therapeutic effects, likely reflecting mechanistic heterogeneity and receptor-related inter-individual differences [105,106]. In affective disorders, insufficient OXT signaling may heighten hypothalamic–pituitary–adrenal axis reactivity and amygdala responsiveness, promoting anxiety and depressive symptoms [107]. Emerging data also implicate OXT in obsessive–compulsive disorder and addiction, where altered cortico–striatal and mesolimbic pathways may contribute to compulsive behaviors and altered reward processing [108].

Neuroimaging studies consistently demonstrate reduced activation in the amygdala, anterior cingulate cortex, and insular cortex in states of low OXT availability, underscoring its central role in socio-affective processing [109]. While these findings provide mechanistic insight, most evidence derives from adult or mixed-age cohorts; pediatric-specific data remain limited, and age-dependent effects require further study.

### 4.3. Endocrine and Growth-Related Features

Although OXT does not directly regulate linear growth, it contributes to the physiological conditions that support healthy development. Through its effects on bone remodeling, body composition, and energy metabolism, OXT-D may indirectly influence somatic maturation in childhood and adolescence. In hypothalamic–pituitary disorders, endocrine alterations often reflect quantitative OXT-D, whereas changes in pubertal timing, stress reactivity, or osmoregulation likely arise from network-level dysregulation. When OXT-D coexists with broader hypothalamic–pituitary dysfunction, its impact may compound other hormonal deficits, affecting growth, metabolism, and neuroendocrine function [39,110].

OXT also interacts with key endocrine axes: it acts with AVP in osmoregulation, modulates adrenocorticotropic hormone-cortisol responses to stress, and may influence gonadotropin secretion and pubertal development [33,111]. Persistent abnormalities in body composition, fatigue, or quality of life in children with hypothalamic–pituitary disorders—despite optimal replacement of classical pituitary hormones—may therefore reflect unrecognized OXT-D [4,39].

### 4.4. Insights from Animal and Translational Models

Translational work in rodents and non-human primates has been essential for clarifying how OXT-D contributes to metabolic, social, and behavioral outcomes [4]. OXT- and OXTR-knockout models reproduce hallmark features seen in humans, including obesity, anxiety-like behavior, social avoidance, and impaired maternal care. Early disruption of oxytocinergic signaling alters hypothalamic and limbic connectivity, producing persistent deficits in sensory processing and socio-emotional learning [112,113,114].

These models also demonstrate both quantitative loss of OXT and functional receptor-level impairment, reinforcing a multidimensional view of OXT-D. Importantly, the oxytocinergic system retains developmental plasticity: reintroduction of OXT or activation of downstream pathways during critical windows can partially reverse social and metabolic abnormalities [115,116,117]. Such findings support the therapeutic potential of OXT-based interventions in pediatric genetic or acquired OXT-D.

## 5. Diagnostic Challenges in Identifying Oxytocin Deficiency

The diagnosis of OXT-D in childhood and adolescence remains particularly challenging due to both methodological and physiological limitations. Diagnostic uncertainty is partly due to the fact that current assays cannot distinguish between quantitative hormone deficiency and functional or network-level OXT-D, which often coexist clinically. Circulating OXT is present at extremely low concentrations, has a plasma half-life of only a few minutes, and follows a pulsatile secretion pattern that complicates interpretation of single measurements [16,118]. Although OXT release increases in response to physiological stimuli such as lactation, parturition, or stress [118,119], capturing secretion dynamics would require frequent sampling and advanced analytical methods—approaches rarely feasible in pediatric practice. Even under controlled conditions, rapid fluctuations contribute to substantial intra- and inter-individual variability [120].

Analytical issues further limit diagnostic accuracy. Conventional enzyme- and radioimmunoassays show poor specificity, cross-reactivity with AVP, and matrix interference. Liquid chromatography–tandem mass spectrometry (LC–MS/MS) provides superior specificity and sensitivity and currently represents the most promising analytical approach for pediatric OXT-D, especially when combined with immunocapture or optimized extraction procedures [121,122]. Its ability to distinguish OXT from structurally related peptides and metabolites also helps overcome the cross-reactivity and matrix interference that limit enzyme-linked immunosorbent assays and radioimmunoassays performance, particularly in pediatric plasma [121,123]. However, this technique remains confined to specialized laboratories and is not standardized for clinical use. Moreover, peripheral OXT levels do not reliably reflect central activity, limiting the clinical value of direct biochemical testing [124].

Provocative testing has been explored as an indirect approach, but no stimulus—including arginine, hypertonic saline, angiotensin II, macimorelin, insulin-induced hypoglycemia, or corticotropin-releasing hormone —has produced reproducible or clinically useful responses [125,126,127,128,129]. Hypertonic saline evokes modest and inconsistent increases in OXT, and insulin-induced hypoglycemia is not suitable for children. A recent protocol using 3,4-methylenedioxymethamphetamine has shown a blunted OXT and neurophysin I response in patients with hypothalamic–pituitary damage, suggesting potential diagnostic value [130,131], but pediatric validation is still lacking.

Additional pediatric challenges include the absence of age-specific reference ranges, lack of validated biomarkers, and ethical concerns surrounding invasive sampling [132]. Neurophysin I appears promising as a more stable peripheral surrogate, but requires further study. At present, diagnosis relies mainly on clinical features—metabolic, psychosocial, and neurobehavioral—within the context of hypothalamic–pituitary disorders [9,133].

Neuroimaging has no direct diagnostic role. Imaging genetics studies, such as those by Xiao et al., link *OXT*-related variants to altered corticostriatal connectivity, but these methods are not applicable to clinical diagnostics. Magnetic resonance imaging remains useful only for detecting structural hypothalamic–pituitary abnormalities [134,135].

Bridging peripheral and central assessments remains a major challenge. Neurophysin I measurement and OXTR methylation profiling are emerging as feasible surrogate approaches, while functional neuroimaging provides mechanistic insight but is not yet practical for routine pediatric assessment. These modalities may eventually help approximate central oxytocinergic function in future research.

### Preliminary Clinical Indicators of Pediatric Oxytocin Deficiency

Although no validated diagnostic criteria currently exist for pediatric OXT-D, emerging evidence allows the identification of a few clinical features that may help orient suspicion. OXT-D is most often characterized by a combination of metabolic abnormalities—such as impaired satiety, reduced energy expenditure, and increased fat mass—and subtle neurobehavioral traits including reduced social engagement, emotional dysregulation or anxiety. Some children may also present with reduced bone mineral density or diminished lean mass despite adequate replacement of other pituitary hormones.

Suspicion increases in specific contexts, particularly in patients with hypothalamic–pituitary tumors, PWS or SYS, or early-life nutritional or environmental adversity. Unlike AVP or ACTH deficiency, OXT-D typically lacks polyuria, polydipsia or electrolyte imbalance, and emotional–social symptoms often persist despite normalization of other hormonal axes.

Peripheral OXT measurements or *OXTR* methylation profiles may offer supportive clues but remain insufficient for diagnosis. These features should therefore be viewed only as early, non-definitive indicators to guide clinical reasoning until standardized diagnostic tools become available.

## 6. Therapeutic Perspectives

### 6.1. Oxytocin Administration Strategies

Therapeutic approaches for OXT-D in children remain limited but are rapidly evolving. Given the multisystem role of OXT across neurobehavioral, metabolic, and endocrine domains, interest in its therapeutic potential has expanded, with an increasing number of pediatric trials now underway [10].

Intranasal administration is the most practical and extensively studied route, allowing partial nose-to-brain delivery via olfactory and trigeminal pathways [136,137,138]. Central availability occurs within 30–45 min, although the proportion reaching target circuits remains uncertain [136,137,139,140]. Because peripheral absorption is minimal, circulating OXT levels do not reliably indicate central activity. Several factors appear to modulate the clinical response to intranasal OXT in pediatric populations. Genetic background—particularly common *OXTR* polymorphisms (e.g., rs53576, rs2254298)—influences neural and behavioral sensitivity to OXT and may partly explain heterogeneous treatment outcomes [141]. Dosing frequency is also important, as intermittent or alternate-day regimens tend to preserve efficacy and reduce tachyphylaxis compared with daily dosing. Pediatric doses typically range from 8 to 48 IU/day, administered once or twice daily, sometimes timed to coincide with socially or emotionally meaningful contexts to enhance behavioral effects [100,142]. Age and developmental stage further shape treatment responsiveness, with younger individuals showing distinct neural connectivity patterns and behavioral effects [143]. Finally, baseline OXT status and the context of administration—especially when paired with structured social or behavioral engagement—appear to enhance therapeutic impact [144].

Systemic routes (intravenous, subcutaneous, oral) remain experimental due to the peptide’s short plasma half-life and limited brain penetration. Novel sustained-release or nanoparticle-based formulations designed to improve stability and central delivery are in early preclinical development [145].

### 6.2. Clinical Trials and Applications in Pediatric Disorders

Clinical trials of intranasal OXT in pediatric disorders have focused primarily on PWS, ASD, and hypothalamic obesity after craniopharyngioma [Table 2].

In PWS, studies suggest benefits for hyperphagia, social behavior, and anxiety, although results remain heterogeneous. Therapeutic response likely reflects the combined contribution of quantitative OXT reduction due to hypothalamic dysfunction and functional impairment of OXTR-mediated social and metabolic pathways. Early-phase trials in infants under six months showed improvements in feeding (including suction and swallowing), social engagement, and motor development, with some evidence of persisting metabolic and endocrine advantages when treatment begins during early developmental windows [146,147]. Trials in older children and adolescents report reductions in hyperphagia and anxiety and better social interaction, although the magnitude of the benefits varies with dose, age, and baseline phenotype [147,148]. Higher doses (32–40 IU/day) may increase irritability or aggression, whereas lower doses (18–24 IU/day) appear better tolerated [149,150]. Overall, OXT shows therapeutic promise in PWS—especially with early initiation—but larger controlled studies are needed to establish efficacy, refine dosing, and determine long-term safety [151,152].

In ASD, outcomes have been mixed. The large phase III trial by Sikich et al. and the multicenter study by Daniels et al. found no significant improvements in social responsiveness with daily intranasal OXT [100,153]. ASD cohorts may primarily exhibit functional OXT-D or network-level dysregulation, which could explain the inconsistent benefits of fixed-dose intranasal OXT in large trials. Conversely, smaller studies have reported improvements in social cognition, motivation, communication, and reductions in repetitive behaviors [154,155]. Baseline biomarkers—including low endogenous OXT levels and specific *OXTR* genotypes—may predict treatment response, supporting precision-medicine approaches [156].

In hypothalamic obesity following craniopharyngioma, small pilot studies indicate that adjunct intranasal OXT may reduce impulsivity, anxiety, and hyperphagia, although consistent improvements in body mass index or metabolic outcomes have not been demonstrated [57,157]. This phenotype likely reflects quantitative OXT-D from hypothalamic injury with additional network-level dysregulation affecting feeding and reward circuits.

**Table 2 cimb-47-00982-t002:** Clinical trials of oxytocin administration in children/adolescents.

	Prader–Willi Syndrome		Autism Spectrum Disorder	Hypothalamic Obesity Post-Craniopharyngioma
**Population**	Infants under six months	Children and adolescents	Children/adolescents 3–18 years	Children and adolescents
**Trial design and size**	Early-phase, small RCT [146,147]	Small RCTs, heterogeneous samples [148,149,150,158,159,160]	Phase III studies and smaller RCTs [37,100,153,155,161,162,163,164]	Pilot RCTs [157]
**Dose and duration (intranasal administration)**	4 IU every other day–8 IU daily.	18–40 IU daily	8–80 IU daily (24 IU in most studies), intermittent dosing; 5–24 weeks	24 IU daily, short term
**Main findings**	Improved feeding (suction, swallowing), social engagement, motor abilities (crawling); benefits are greater with early treatment	Reduced hyperphagia, anxiety, and improved social behaviors; higher doses or older age associated with irritability/aggression; personalized dosing recommended	Mixed efficacy; the largest Phase III trial showed no significant benefit; subgroup benefits in younger or intellectually disabled children; biomarkers (low baseline OXT, OXTR genotype) may predict response	Reduced impulsivity, anxiety, hyperphagia; no significant body-mass-index or metabolic improvements

Abbreviations: IU, international units; OXT, oxytocin; OXTR, oxytocin receptor; RCT, randomized controlled trial.

### 6.3. Safety and Efficacy Considerations

Across pediatric studies, intranasal OXT shows a favorable short-term safety profile. Reported adverse events—including nasal irritation, mild sedation, or transient headache—are generally mild and self-limited, and no consistent changes in vital signs, laboratory parameters, or body mass index have been documented. However, long-term data are scarce, and the effects of chronic exposure on neurodevelopment, receptor sensitivity, or socio-emotional functioning remain insufficiently understood [9,100,147,156].

Efficacy results are highly variable across trials, reflecting heterogeneity in patient characteristics, dosing schedules, and behavioral endpoints. Genetic factors (e.g., *OXTR* and *CD38* polymorphisms), baseline OXT concentrations, and the psychosocial context of administration appear to modulate treatment response, while continuous daily dosing may promote receptor desensitization and reduce effectiveness over time [154,156]. A recent meta-analysis by Zhang et al. indicates that baseline OXT levels may help identify more responsive subgroups. These data support the adoption of personalized dosing strategies and the integration of structured behavioral interventions to optimize therapeutic benefit [156].

### 6.4. Future Directions

Future therapeutic development for pediatric OXT-D will increasingly emphasize precision and personalization. Novel delivery systems—including sustained-release intranasal formulations, nanoparticle-encapsulated sprays, and liposomal carriers—aim to enhance central bioavailability and extend duration of action [145]. Combination treatments targeting complementary pathways, such as growth hormone or leptin modulation, are also being explored for complex hypothalamic–pituitary disorders [75,165].

Beyond intranasal preparations, several emerging modalities may offer advantages in the long term. Long-acting OXT analogs with improved pharmacokinetics show superior brain penetration and sustained behavioral effects in preclinical models [166]. Selective OXTR agonists and OXT-derived peptides are under investigation for prosocial and metabolic actions, although human data remain limited [167,168]. Epigenetic and gene-targeting strategies aimed at modulating OXTR expression represent an additional frontier for durable modulation of the oxytocinergic system [169]. While none of these approaches are yet applicable to clinical pediatric care, they highlight important avenues for future therapeutic innovation.

A major priority is the development of biomarker-driven and genotype-informed approaches to identify children most likely to benefit from therapy. Integration of neuroimaging and molecular markers—such as baseline OXT concentrations, *OXTR* methylation profiles, and resting-state connectivity within OXT-related networks—may support individualized decision-making. Insights from OXT-knockout and OXTR-epigenetic models continue to guide translational progress and inform the design of future pediatric trials, particularly those incorporating biomarker-based stratification and age-specific endpoints.

Precision-medicine strategies are also emerging for syndromic OXT-D, particularly in PWS and SYS. Early-life intervention during sensitive neurodevelopmental windows appears especially promising, supported by Magel2-deficient models and early clinical trials in infants with PWS. Stratification by genotype (e.g., deletion vs. uniparental disomy), phenotype (including sex differences and ASD-like traits), and neurobehavioral profiles may help identify the most responsive subgroups. Preliminary evidence also suggests that OXTR polymorphisms, epigenetic markers, and neuroimaging signatures could refine patient selection, though none are yet ready for routine clinical use.

Large, multicenter, long-term longitudinal studies remain essential to define optimal timing, dosing, and safety, and to clarify the long-term metabolic, endocrine, and neuropsychiatric effects of OXT-based therapies in pediatric populations.

## 7. Conclusions

OXT-D is increasingly recognized as a clinically relevant condition in pediatric endocrinology, especially in hypothalamic–pituitary disorders and selected genetic syndromes. Rather than a single entity, OXT-D encompasses a spectrum that includes quantitative hormone deficiency, functional receptor impairment, and broader network-level dysregulation. Its clinical manifestations span metabolic alterations, impaired body composition, and socio-emotional and neurobehavioral dysfunctions, reflecting the integrative role of OXT across development. However, diagnosis remains difficult due to pulsatile secretion, limited assay reliability, and the absence of standardized criteria or validated biomarkers. Intranasal OXT shows promise, though therapeutic responses are heterogeneous and long-term safety and efficacy remain uncertain.

Future work should prioritize the development of reliable diagnostic tools, biomarker-guided patient stratification, and rigorously designed longitudinal trials to define optimal timing, dosing, and treatment context. Given the multifaceted phenotype of OXT-D, early recognition in high-risk groups and coordinated multidisciplinary management are essential. Continued translational and clinical research will be critical for clarifying therapeutic potential and improving outcomes for children and adolescents affected by this emerging neuroendocrine disorder.


## Figures and Tables

**Table 1 cimb-47-00982-t001:** Congenital and acquired causes of pediatric oxytocin deficiency.

	Etiology	Pathophysiological Mechanism	Clinical Correlates
**Congenital causes**	Prader–Willi syndrome	Loss of paternally expressed genes on 15q11–q13 → impaired development and signaling of hypothalamic OXT neurons	Neonatal hypotonia, hyperphagia, obesity, and emotional dysregulation
	Schaaf–Yang syndrome (*MAGEL2* variants)	Altered neurodevelopment and OXT neuron differentiation, similar to PWS	Hypotonia, feeding difficulties, autism-like features, and social dysfunction
	Williams syndrome	Dysregulated OXT/AVP pathways in neurodevelopmental circuits	Hypersociability, anxiety, impaired social cognition
	Fragile X syndrome	Synaptic and neuropeptide dysregulation impacting OXT/AVP release	Social anxiety, cognitive delay, autistic traits
	Autism spectrum disorder	OXTR gene polymorphisms and promoter methylation → reduced receptor expression and oxytocinergic tone	Impaired social interaction, communication deficits
**Acquired causes**	Craniopharyngioma	Tumor and/or treatment-induced injury to paraventricular and supraoptic nuclei	Blunted OXT response to stimuli, social and emotional dysfunction
	Germ cell tumors, hypothalamic gliomas, Langerhans cell histiocytosis	Structural damage to hypothalamic nuclei	Secondary neuroendocrine deficits, behavioral changes
	Autoimmune/inflammatory hypophysitis (lymphocytic, IgG4-related)	Immune-mediated hypothalamic–posterior pituitary injury	Possible secondary OXT loss; anxiety, altered stress responses
	Traumatic brain injury or neurosurgical lesions	Disruption of axonal transport and neuronal connectivity in OXT pathways	Emotional lability, altered feeding and social regulation
	Secondary neurohypophyseal dysfunction (central diabetes insipidus, panhypopituitarism)	Involvement of the posterior pituitary adjacent to OXT-secreting neurons	Variable OXT deficiency, often subclinical; impaired socio-emotional adaptation
**Genetic and epigenetic causes**	*OXTR* gene polymorphisms (e.g., rs53576, rs2254298)	Altered receptor expression, ligand affinity, and signaling efficiency	Variability in social cognition, stress response, attachment behaviors, and autism spectrum disorder features
	*OXTR* epigenetic modifications (DNA methylation, histone modification)	Reduced OXTR expression due to promoter methylation and environmental influences (stress, malnutrition)	Lower peripheral OXT, diminished social sensitivity, and emotional dysregulation
**Nutritional factors**	Prenatal and early-life undernutrition	Disrupted hypothalamic OXT and OXTR expression via developmental reprogramming	Increased fat mass, reduced lean body mass, and altered metabolic and emotional regulation
	Chronic energy deprivation (e.g., anorexia nervosa, oligomenorrheic athletes)	Suppressed OXT secretion as an adaptive response to conserve energy	Low OXT correlates with leptin reduction, low bone density, and impaired socio-emotional functioning

Abbreviations: AVP, vasopressin; OXT, oxytocin; OXTR, oxytocin receptor.

## Data Availability

No new data were created or analyzed in this study. Data sharing is not applicable to this article.

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
