# Peer review of "Oxytocin Deficiency in Childhood and Adolescence: Clinical Features, Diagnostic Challenges and Therapeutic Perspectives"

_cimb, 2025, doi:10.3390/cimb47120982_

Round 1
Reviewer 1 Report
Comments and Suggestions for Authors
Article " Oxytocin Deficiency in Childhood and Adolescence: Clinical Features, Diagnostic Challenges and Therapeutic Perspectives" by Roberto Paparell et al. This is a well-written and clearly structured narrative review that tackles a genuinely emerging and clinically relevant concept: oxytocin deficiency (OXT-D) as a distinct pediatric neuroendocrine entity, especially in hypothalamic–pituitary and genetic syndromes. The manuscript provides a comprehensive overview of oxytocin physiology, the developmental aspects of OXT and vasopressin systems, and the multifactorial etiology of OXT-D in childhood and adolescence, spanning genetic, structural, nutritional, and epigenetic causes. Details clinical manifestations across metabolic, neurobehavioral, psychiatric, and endocrine domains. Focusing on diagnostic challenges and the current state of therapeutic interventions, with emphasis on intranasal OXT in Prader–Willi syndrome (PWS), autism spectrum disorder (ASD), and hypothalamic obesity after craniopharyngioma. The article is well written in a clear and understandable language, the conclusions are logical, the literature corresponds to the stated topic.
Some comments:
1) At present, OXT-D is used to cover at least three partially overlapping but different entities: true quantitative deficiency of OXT secretion (for example after hypothalamic damage), functional deficiency due to impaired receptor availability or signaling (for example OXTR polymorphisms or promoter methylation), and broader “dysregulation” or imbalance within OXT/AVP networks or downstream circuits. I strongly recommend that the authors define early on what they mean by OXT-D, distinguish “hormone deficiency” from “impaired system function” and “context-dependent dysregulation,” and then adhere to those definitions.
2) For readers of a pediatric-focused review, it would be very useful if you would consistently indicate, even briefly, whether a given study was in children, adolescents, adults, or mixed cohorts. For instance, in the metabolic and bone subsections, you rely quite heavily on adult or young adult data to infer pediatric relevance; that is acceptable, but it should be transparent. Similarly, some neuropsychiatric associations (schizophrenia, OCD, addiction) are largely adult, and their inclusion in a pediatric-centered review should be clearly framed as extrapolations with limited direct pediatric evidence.
3) Structurally, the manuscript is long (27 pages) and occasionally repetitive. Concepts such as “pleiotropic hormone,” “integrative role at the interface of metabolism and social behavior,” and “emerging clinical entity” recur with similar wording in the Introduction, Physiological sections, Clinical manifestations, and Conclusions. A careful linguistic tightening could shorten the text without losing content and would make it more readable. For example, the description of OXT’s roles in metabolism, bone, and muscle appears in some form in the Abstract, Introduction, sections 2.1–2.4, 4.1, and 4.3. You might consider consolidating some of this into one definitive paragraph and a single integrative figure, and then referring back to it rather than rephrasing multiple times.
I recommend that the authors discuss the above mentioned points in the text of the article.
Overall, the study is comprehensive and can be recommended for publication.
Author Response
We thank the reviewers for the careful reading of the manuscript and the constructive remarks. We believe they have helped us to substantially improve our manuscript. We provided a detailed point-by-point response to all comments in order to improve and clarify the manuscript. All changes in the revised manuscript are highlighted in light yellow (rev 1) and green (rev 2). Reviewers’ original comments are listed below, followed by our response to each comment.
|
Point-by-Point Response to Comments and Suggestions for Authors |
Article "Oxytocin Deficiency in Childhood and Adolescence: Clinical Features, Diagnostic Challenges and Therapeutic Perspectives" by Roberto Paparella et al. This is a well-written and clearly structured narrative review that tackles a genuinely emerging and clinically relevant concept: oxytocin deficiency (OXT-D) as a distinct pediatric neuroendocrine entity, especially in hypothalamic–pituitary and genetic syndromes. The manuscript provides a comprehensive overview of oxytocin physiology, the developmental aspects of OXT and vasopressin systems, and the multifactorial etiology of OXT-D in childhood and adolescence, spanning genetic, structural, nutritional, and epigenetic causes. Details clinical manifestations across metabolic, neurobehavioral, psychiatric, and endocrine domains. Focusing on diagnostic challenges and the current state of therapeutic interventions, with emphasis on intranasal OXT in Prader–Willi syndrome (PWS), autism spectrum disorder (ASD), and hypothalamic obesity after craniopharyngioma. The article is well written in a clear and understandable language, the conclusions are logical, and the literature corresponds to the stated topic.
Some comments:
We sincerely thank the Reviewer for the constructive and insightful comments. We appreciate the recognition of the manuscript’s novelty and interdisciplinary approach. Below, we provide a point-by-point response and indicate the revisions incorporated into the manuscript.
- At present, OXT-D is used to cover at least three partially overlapping but different entities: true quantitative deficiency of OXT secretion (for example, after hypothalamic damage), functional deficiency due to impaired receptor availability or signaling (for example,e OXTR polymorphisms or promoter methylation), and broader “dysregulation” or imbalance within OXT/AVP networks or downstream circuits. I strongly recommend that the authors define early on what they mean by OXT-D, distinguish “hormone deficiency” from “impaired system function” and “context-dependent dysregulation,” and then adhere to those definitions.
We thank the reviewer for this important conceptual clarification. In the revised manuscript, we have now introduced an explicit definition of “oxytocin deficiency (OXT-D)” in the Introduction. Specifically, we differentiate among:
(1) quantitative deficiency
(2) functional deficiency
(3) context-dependent dysregulation within the OXT/AVP system or downstream circuits.
We also revised the subsequent sections to ensure terminological consistency and to avoid using “OXT-D” ambiguously when discussing receptor-level or circuit-level dysfunction. Where appropriate, we now specify “impaired oxytocin signaling” or “oxytocin system dysregulation” rather than “deficiency.”
- For readers of a pediatric-focused review, it would be very useful if you would consistently indicate, even briefly, whether a given study was in children, adolescents, adults, or mixed cohorts. For instance, in the metabolic and bone subsections, you rely quite heavily on adult or young adult data to infer pediatric relevance; that is acceptable, but it should be transparent. Similarly, some neuropsychiatric associations (schizophrenia, OCD, addiction) are largely adult, and their inclusion in a pediatric-centered review should be clearly framed as extrapolations with limited direct pediatric evidence.
We appreciate this recommendation and fully agree that clarity on population age is essential for a pediatric-focused review. We have revised the manuscript accordingly.
- Structurally, the manuscript is long (27 pages) and occasionally repetitive. Concepts such as “pleiotropic hormone,” “integrative role at the interface of metabolism and social behavior,” and “emerging clinical entity” recur with similar wording in the Introduction, Physiological sections, Clinical manifestations, and Conclusions. A careful linguistic tightening could shorten the text without losing content and would make it more readable. For example, the description of OXT’s roles in metabolism, bone, and muscle appears in some form in the Abstract, Introduction, sections 2.1–2.4, 4.1, and 4.3. You might consider consolidating some of this into one definitive paragraph and a single integrative figure, and then referring back to it rather than rephrasing multiple times.
We have carefully revised the manuscript to address this suggestion.
I recommend that the authors discuss the above mentioned points in the text of the article.
Overall, the study is comprehensive and can be recommended for publication.
We thank the Reviewer for the comments. We have addressed the abovementioned points in the revised version of the manuscript.
Reviewer 2 Report
Comments and Suggestions for Authors
The author reported in this review article “Oxytocin Deficiency in Childhood and Adolescence: Clinical Features, Diagnostic Challenges and Therapeutic Perspectives” The present review describes a thorough and insightful synthesis of current knowledge on pediatric oxytocin deficiency (OXT-D), effectively underscoring its clinical significance in children with hypothalamic-pituitary disorders and genetic syndromes such as Prader-Willi and Schaaf-Yang. The authors successfully integrate evidence from both experimental models and human studies to illustrate how OXT-D contributes to metabolic dysregulation, altered body composition, impaired bone health, and socio-emotional challenges, even in the context of adequate replacement of other pituitary hormones.
Importantly, the review highlights the diagnostic difficulties stemming from OXT’s short plasma half-life, pulsatile secretion, and the methodological shortcomings of existing assays, though a more in-depth critique and comparison of these analytical techniques would strengthen this section. The discussion of intranasal OXT therapy is balanced and informative, detailing its safety, variable efficacy, and context-dependent behavioral effects; however, expanding on factors influencing treatment responsiveness-such as developmental timing, genotype, or dosing paradigms would enhance clinical relevance.
The authors’ call to recognize OXT-D as a distinct clinical entity is well justified, but the manuscript would benefit from a clearer framework outlining potential diagnostic criteria, proposed biomarkers, and practical considerations for clinical implementation. Overall, this is a timely and well-organized review that advances the understanding of pediatric OXT-D, and with targeted refinements, it will serve as a valuable reference for both clinicians and researchers in pediatric endocrinology and neurodevelopment. In my opinion, this paper would be a nice contribution to the readers of Current Issues in Molecular Biology (CIMB-MDPI). I will make some suggestions that may improve the reach of the paper (in terms of reaching a broad audience). As the modifications can be addressed in straight forward manner and I am recommending for the publication of this article after the following recommended changes in the manuscript:
- Could the authors clarify which specific clinical or biochemical features they propose should be included in preliminary diagnostic criteria for pediatric OXT-D, and how these might be differentiated from overlapping hypothalamic–pituitary deficiencies?
- The manuscript mentions significant limitations in current OXT-assays. Can the authors provide more detail on which assay platforms (e.g., RIA, ELISA, LC–MS/MS) show the greatest promise for improving diagnostic accuracy in children?
- How do the authors envision bridging the gap between peripheral OXT measurements and central oxytocinergic activity? Are there emerging surrogate biomarkers or imaging modalities that could be clinically feasible?
- Given the modest and context-dependent outcomes of intranasal OXT therapy, can the authors elaborate on the factors that most strongly influence treatment response (e.g., age, genetic background, baseline OXT status, dosing frequency)?
- Are there ongoing longitudinal studies assessing the long-term metabolic, behavioral, or neurodevelopmental effects of chronic intranasal OXT use in pediatric populations?
- For conditions such as Prader–Willi and Schaaf–Yang syndromes, do the authors recommend specific screening timelines or monitoring strategies for early detection of OXT-D?
- Could the authors expand on what precision-medicine strategies they envision for OXT-D management, and whether stratification based on genotype, phenotype, or neurobehavioral profile is feasible?
- Aside from intranasal OXT, are there any emerging therapeutic modalities (e.g., OXT analogs, receptor agonists, gene therapy approaches) that the authors believe hold promise for pediatric application?
- The review article notes challenges with dynamic testing. Are there any proposed stimulation or suppression tests under investigation that might offer improved diagnostic utility?
- Which experimental models or clinical trial designs do the authors consider most critical to advancing the understanding of OXT-D pathophysiology and treatment in children?
Author Response
We thank the reviewers for the careful reading of the manuscript and the constructive remarks. We believe they have helped us to substantially improve our manuscript. We provided a detailed point-by-point response to all comments in order to improve and clarify the manuscript. All changes in the revised manuscript are highlighted in light yellow (rev 1) and green (rev 2). Reviewers’ original comments are listed below, followed by our response to each comment.
|
Point-by-Point Response to Comments and Suggestions for Authors |
The author reported in this review article “Oxytocin Deficiency in Childhood and Adolescence: Clinical Features, Diagnostic Challenges and Therapeutic Perspectives” The present review describes a thorough and insightful synthesis of current knowledge on pediatric oxytocin deficiency (OXT-D), effectively underscoring its clinical significance in children with hypothalamic-pituitary disorders and genetic syndromes such as Prader-Willi and Schaaf-Yang. The authors successfully integrate evidence from both experimental models and human studies to illustrate how OXT-D contributes to metabolic dysregulation, altered body composition, impaired bone health, and socio-emotional challenges, even in the context of adequate replacement of other pituitary hormones.
Importantly, the review highlights the diagnostic difficulties stemming from OXT’s short plasma half-life, pulsatile secretion, and the methodological shortcomings of existing assays, though a more in-depth critique and comparison of these analytical techniques would strengthen this section. The discussion of intranasal OXT therapy is balanced and informative, detailing its safety, variable efficacy, and context-dependent behavioral effects; however, expanding on factors influencing treatment responsiveness-such as developmental timing, genotype, or dosing paradigms would enhance clinical relevance.
The authors’ call to recognize OXT-D as a distinct clinical entity is well justified, but the manuscript would benefit from a clearer framework outlining potential diagnostic criteria, proposed biomarkers, and practical considerations for clinical implementation. Overall, this is a timely and well-organized review that advances the understanding of pediatric OXT-D, and with targeted refinements, it will serve as a valuable reference for both clinicians and researchers in pediatric endocrinology and neurodevelopment. In my opinion, this paper would be a nice contribution to the readers of Current Issues in Molecular Biology (CIMB-MDPI). I will make some suggestions that may improve the reach of the paper (in terms of reaching a broad audience). As the modifications can be addressed in straight forward manner and I am recommending for the publication of this article after the following recommended changes in the manuscript:
We thank the Reviewer for the careful reading of our manuscript and for the constructive and encouraging comments. We appreciate the recognition of the clinical relevance and the comprehensive nature of our review, and we agree with the reviewer’s suggestions. Below we provide a point-by-point response and indicate the revisions incorporated into the manuscript.
- Could the authors clarify which specific clinical or biochemical features they propose should be included in preliminary diagnostic criteria for pediatric OXT-D, and how these might be differentiated from overlapping hypothalamic–pituitary deficiencies?
We thank the reviewer for this important and clinically relevant question. In response, we have now added a short subsection within the Diagnostic Challenges section (5.1) proposing a set of preliminary, non-validated diagnostic criteria for pediatric OXT-D.
- The manuscript mentions significant limitations in current OXT-assays. Can the authors provide more detail on which assay platforms (e.g., RIA, ELISA, LC–MS/MS) show the greatest promise for improving diagnostic accuracy in children?
We have now added a short paragraph regarding assay platforms within section 5.
- How do the authors envision bridging the gap between peripheral OXT measurements and central oxytocinergic activity? Are there emerging surrogate biomarkers or imaging modalities that could be clinically feasible?
The gap between peripheral OXT levels and central oxytocinergic activity may be addressed through emerging surrogate biomarkers such as neurophysin I, which provides a more stable and reliable readout of OXT secretion, and through epigenetic markers such as OXTR methylation. Functional neuroimaging offers mechanistic insight but is not yet feasible for routine pediatric use. These approaches represent the most promising strategies to approximate central OXT function in clinical research.
- Given the modest and context-dependent outcomes of intranasal OXT therapy, can the authors elaborate on the factors that most strongly influence treatment response (e.g., age, genetic background, baseline OXT status, dosing frequency)?
We thank the reviewer for this important point. We have now added a dedicated paragraph summarizing the main factors influencing treatment response to intranasal OXT in children—including OXTR genotype, dosing frequency, age, baseline OXT status, and contextual variables—and clarified how these determinants may contribute to heterogeneous clinical outcomes (see Therapeutic Perspectives section).
- Are there ongoing longitudinal studies assessing the long-term metabolic, behavioral, or neurodevelopmental effects of chronic intranasal OXT use in pediatric populations?
We thank the reviewer for this important question. To date, no large, long-term longitudinal trials have evaluated the chronic metabolic, behavioral, or neurodevelopmental consequences of intranasal OXT in pediatric populations. Only short-term or medium-duration studies (typically 1–24 weeks) are available, and ongoing registered trials focus primarily on short-term behavioral outcomes in ASD and PWS. We have highlighted the need for longitudinal, multicenter studies investigating long-term safety and developmental effects.
- For conditions such as Prader–Willi and Schaaf–Yang syndromes, do the authors recommend specific screening timelines or monitoring strategies for early detection of OXT-D?
At present, we do not recommend syndrome-specific screening timelines for early detection of OXT-D in Prader–Willi or Schaaf–Yang syndromes. This is mainly because no validated biochemical test exists, oxytocin assays show high variability and poor clinical correlation, and current clinical guidelines for these syndromes do not include OXT measurement. Early monitoring in these conditions should therefore remain focused on feeding abnormalities, growth trajectories, hypotonia, and socio-behavioral development—features that may reflect underlying OXT-system dysfunction but are not specific enough to justify targeted screening. Given the absence of consensus on laboratory markers or reliable diagnostic thresholds, routine OXT screening cannot yet be recommended outside research settings.
- Could the authors expand on what precision-medicine strategies they envision for OXT-D management, and whether stratification based on genotype, phenotype, or neurobehavioral profile is feasible?
Thank you for this insightful comment. We have expanded the manuscript to clarify emerging precision-medicine strategies for OXT-D management. Early intervention during critical neurodevelopmental windows, together with stratification based on genotype (e.g., MAGEL2 or PWS deletion subtypes), phenotype (such as sex and ASD-like traits), and neurobehavioral profiles, appears feasible and increasingly supported by both clinical and preclinical studies. We also discuss how OXTR polymorphisms, epigenetic markers, and neuroimaging may contribute to future personalized approaches, although these tools are not yet ready for routine clinical use.
- Aside from intranasal OXT, are there any emerging therapeutic modalities (e.g., OXT analogs, receptor agonists, gene therapy approaches) that the authors believe hold promise for pediatric application?
We thank the reviewer for this insightful suggestion. Although evidence is currently limited to preclinical or early translational models, these approaches show potential for longer-lasting metabolic and neurobehavioral effects. We have added a concise overview of these modalities to the manuscript.
- The review article notes challenges with dynamic testing. Are there any proposed stimulation or suppression tests under investigation that might offer improved diagnostic utility?
We thank the reviewer for this comment. No stimulation or suppression test is currently established for clinical diagnosis of OXT-D in pediatric populations, but several provocative tests are under investigation. The MDMA stimulation test with measurement of oxytocin and neurophysin I is the leading candidate for improved diagnostic utility in OXT-D, but further validation, especially in pediatric and therapeutic contexts, is required before clinical adoption.
- Which experimental models or clinical trial designs do the authors consider most critical to advancing the understanding of OXT-D pathophysiology and treatment in children?
The most informative experimental models for understanding pediatric OXT-D are oxytocin-knockout and OXTR-targeted epigenetic models, which reproduce core social, behavioral, and neurodevelopmental features relevant to clinical pediatric phenotypes. These models have been essential for clarifying the mechanistic consequences of oxytocin loss.
Regarding clinical research, the most critical designs are randomized, double-blind, placebo-controlled trials that stratify participants by age, baseline oxytocin levels, and OXTR genotype, and that incorporate standardized behavioral and physiological endpoints. Recent trials adopting intermittent dosing schedules, controlled social contexts, and extended follow-up periods offer the best insights into efficacy and safety in children.
Although provocative tests (e.g., CRH, MDMA) are being explored, none are yet reliable for diagnostic purposes. Overall, progress will depend on integrative approaches combining robust animal models, biomarker-informed stratification, and rigorously designed pediatric trials.